# Nanoscale Ion-Exchange Materials: From Analytical Chemistry to Industrial and Biomedical Applications

**DOI:** 10.3390/molecules28186490

**Published:** 2023-09-07

**Authors:** Magdalena Matczuk, Lena Ruzik, Bernhard K. Keppler, Andrei R. Timerbaev

**Affiliations:** 1Faculty of Chemistry, Warsaw University of Technology, 00-664 Warsaw, Poland; magdalena.matczuk@pw.edu.pl; 2Institute of Inorganic Chemistry, University of Vienna, 1090 Vienna, Austria; andrei.timerbaev@univie.ac.at

**Keywords:** NIEs, preparation, ion-exchange properties, chemical analysis, industrial applications, drug delivery

## Abstract

Nano-sized ion exchangers (NIEs) combine the properties of common bulk ion-exchange polymers with the unique advantages of downsizing into nanoparticulate matter. In particular, being by nature milti-charged ions exchangers, NIEs possess high reactivity and stability in suspensions. This brief review provides an introduction to the emerging landscape of various NIE materials and summarizes their actual and potential applications. Special attention is paid to the different methods of NIE fabrication and studying their ion-exchange behavior. Critically discussed are different examples of using NIEs in chemical analysis, e.g., as solid-phase extraction materials, ion chromatography separating phases, modifiers for capillary electrophoresis, etc., and in industry (fuel cells, catalysis, water softening). Also brought into focus is the potential of NIEs for controlled drug and contrast agent delivery.

## 1. Introduction

Ion-exchange polymers (IEPs) are synthetic materials that contain active, electrically charged groups along the polymer backbone to form a medium for ion exchange. When cross-linked via copolymerization with a suitable organic linker, the IEPs are converted into 3D structures with improved structural integrity and robustness, which are often called ion-exchange resins. Together with inorganic ion exchangers and natural ion-exchange materials, e.g., zeolites, clays, or soil humus, IEPs comprise a broad class of ion exchangers [1]. The enormous economic impact of IEPs is largely due to their wide use in the hydrometallurgical, chemical and petrochemical, and metal-finishing industries [2,3,4,5], among many others, especially where separation, purification, or decontamination is essential [6,7,8,9,10].

In recent years, we have observed research in both academic and industrial domains on downsizing various materials to have intensified, and this trend has included IEPs. The interest in reducing IEPs down to particles with different morphologies, whose size in at least one dimension is below 100 nm, is driven by the desire to improve the performance of these polymers. However, attracting much more interest is the ability of nano-sized ion exchangers (NIEs) to introduce new nanoscale properties. Among these are large specific surface area and high ion-exchange activity, surface-to-volume ratios, and water dispersity, as well as longer circulation times in biosystems, etc. Furthermore, due to their intrinsic composition, NIEs behave like multi-charged ions, promoting electrostatic repulsion to maintain high stability in suspension with no need of surface modification. All of these attributes have inspired both fundamental research and research activities toward diverse applications. Some applications require NIEs to be mixed or conjugated with other materials, e.g., magnetite [11] or silica gel [12], within a nanoparticle, and these hybrid nanostructures are also reviewed in this paper. On the other hand, beyond its scope, are nanocomposites made primarily of IEPs doped with inorganic nanoparticles [13,14,15]. It should be mentioned that the family of ionic polymer nanomaterials is distinctly larger, including, for instance, an interesting class of nanoparticles based on poly(ionic liquids) [16,17,18], ionically assembled from ionic liquids [19] or grafted with them [20].

To stay on message, this review describes only NIEs in greater detail. We first briefly introduce rational design strategies and then focus on the fabrication procedures, which have a significant effect on each intended application, including their merits, limitations, and complementarity for large-scale production. Also critically discussed are the characterization techniques that enable researchers to better understand the ion-exchange behavior of NIEs, particularly at the nanoscale. Next, the application potential and recent developments of NIEs in the fields of analytical chemistry, nanotechnology, and biomedicine are presented in detail. A summary of the challenges of and future perspectives on NIEs follows to give context to the state of the art of this engaging research field (Table 1).

## 2. Rational Design and Fabrication Strategies

Classical IEPs contain acidic or basic functionalities such as sulfonic and carboxylic acid groups or quaternary ammonium groups, respectively. These are introduced into the polymer matrix, consisting most commonly of cross-linked polystyrene (PS). While NIEs do not differ from IEPs in their functional, ionizable groups, the polymer substrates may be varied from poly(methyl methacrylate) (PMMA) and its derivatives to polyesters and poly(lactic acid), which are generally recognized as safe polymers (Table 1), to boost their application possibilities and potential. In addition, a number of NIEs are based on natural polymer materials such as chitosan [25,40,46], alginate [47], or gelatin [48]. Of these NIEs, chitosan possesses perhaps the best combination of properties that makes it favorable for a plethora of applications [49,50,51], some of which are discussed below (see Section 4). Another inexpensive and easy-to-produce alternative to synthetic NIEs is halloysite, a natural clay material with a nanotubular structure [52] which can be converted into effective cation NIEs in one of two ways (Figure 1; see also [53]).

There are numerous strategies to produce synthetic NIEs with the desired nanoscale characteristics. According to the common classification [54], they can be divided into direct polymerization and post-polymerization methods.

### 2.1. Direct Polymerization

The prevailing method is the direct approach using emulsion or precipitation polymerization, which is associated with nanoparticle functionalization (unless styrene sulfonate is used as an emulsifying comonomer to form an ionic charge on the surface of the nanoparticles in situ [21]). In another (out of the very few) one-step approach for the preparation of NIEs, monomers of 2-(dimethylamino) ethyl methacrylate (or 4-vynylpyridine) and 4-vinylbenzyl chloride are polymerized in the presence of azobisisobutyronitrile and undergo quarterization simultaneously [45]. For fabricating core–shell architectures, e.g., with a core of PMMA and an anionic shell of modified poly(*N*-isopropylacrylamide), a two-stage emulsion polymerization technique is employed, importantly, in aqueous medium [24]. Copolymerization with acrylic or methacrylic acid, each possessing a different reactivity toward the shell polymer, results in the formation of nanoparticles with the same total charge but a dissimilar distribution of carboxylic groups across the shell, either uniformly distributed throughout or with a carboxylic-rich inner shell, respectively (Figure 2).

As mentioned above, in contrast to nonionic polymeric nanoparticles, NIEs can rarely be obtained in one step, requiring a further reaction in order to obtain an ion-exchanging functionality. For this purpose, a sulfonated sodium styrene comonomer was incorporated in the shell of PS latex particles in a two-step seeded polymerization reaction [22]. A score of NIEs were thus produced either with the same size and different surface charge densities or with varying size and a similar surface charge. Likewise, a positively charged unit was introduced into otherwise uncharged diblock poly(stearyl methacrylate)–poly(benzyl methacrylate) nanoparticles to obtain NIEs with a range of morphologies (spheres, worms, and vesicles) [39]. Some researchers gave preference to more tedious synthetic schemes such as incorporating the synthesis of poly(styrene-co-maleic anhydride), its grafting with melamine, and, finally, functionalization by 1,2-ethlynediamine or 1,3-propylenediamine, all steps under the action of ultrasonic irradiation and magnetic stirring to generate nanoscale polymeric particles [55,56,57]. 

### 2.2. Post-Polymerization

When using post-polymerization, polymerization and nanoparticle formation are independent steps. For instance, a series of polyethylene-based NIEs with periodic sulfonate groups was synthesized by polyesterification of octatetraconane-1,48-diol and tetrabutylammonium dimethyl sulfosuccinate, which, after the replacement of the NBu_4_^+^ counterions by Na^+^ or Cs^+^, were transformed into platelet-like self-stabilized NIEs by ultrasonication [38]. Reisch et al. [26] proposed a simple and flexible protocol for the preparation of ultrasmall NIEs based on charge-controlled nanoprecipitation that basically only requires dissolving a charged polymer in a water-miscible solvent and adding this solution to water. In some cases, the assembly of NIEs needs virtually no polymerization, such as for particles that consist of a hydrophobic polymeric core and an anionic methacrylate-based layer, and nanoization can be attained using flash nanoprecipitation [42]. In this kinetically controlled process, an organic stream containing core and amphiphilic polymer materials is mixed under turbulent flow conditions with an aqueous stream, promoting uniform nucleation and particle growth. As a continuous procedure, flash nanoprecipitation makes the scalable production of NIEs feasible. However, greener conditions than using tetrahydrofuran as a solvent in the organic stream would be highly desirable. 

### 2.3. Non-Synthetic Methodology

An alternative “bottom-down” approach was proposed by Dolgonosov and Khamizov [58], who prepared NIEs not synthetically but by the grinding of commercial ion-exchange materials (based on a styrene–divinylbenzene copolymer matrix). To downsize such ion exchangers, the authors used their treatment in a ball mill, followed by prolonged sedimentation and centrifugation of an aqueous suspension of the ground substance. As shown in Table 1 (see also Section 4 for more detail), the obtained NIEs possess a range of attractive properties, enabling their application in various fields. However, the fabrication is complicated by the rather tedious and time-consuming process (more than 100 h of grinding time), resulting in a negligible end-product yield. Also critically, the NIEs, particularly of the anion-exchange type, form colloids with considerable size dispersity, and the size of individual particles may reach 300 nm, thus greatly surpassing the characteristic nanoscale limit.

## 3. Characterization

As with every newly synthesized nanomaterial, NIEs need characterization with regard to their morphology, size, composition and functionalization, stability in suspension, and other nanoscale attributes. However, the arsenal of methods used to acquire these data, such as electron microscopy, dynamic light or small-angle X-ray scattering, ζ potential measurements, and Fourier-transform IR spectroscopy, etc., does not differ from that employed for studying other nanostructures. In view of the potential applications of NIEs, it is of greater interest to consider the parameters governing their ion-exchange behavior (the number of charged groups in the particle shell, total charge and charge distribution, the IEC, etc.) and some other pertinent properties (e.g., conductivity, metal adsorption capacity, the electrophoretic response), as well as the related methodological aspects and toolbox available for the respective measurements.

Surface charge density, i.e., the charge per unit surface area, is an important characteristic of NIEs which can be routinely determined by conductometric titration [22]. The same technique is suitable to determine the molar amount of ion-exchange sites per g of NIE. From the potentiometric titration results, it is possible to evaluate the acid capacity of the HSO_3_-functionalized NIEs (in mmol H^+^ per g) [53]. For determining the IEC or the ability of an NIE to undergo displacement of ions, manually performed acid–base titration is still the method of choice [21,31,40]. Moreover, units used to express this parameter vary greatly, from mmol g^−1^ to meq mL^−1^ or meq g^−1^, which makes a comparison of different NIE systems difficult. On the other hand, the charge density measurements can be carried out instrumentally by a particle charge detector [41]. Another parameter indicative of the number of ionizable groups is the degree of neutralization (or percent of neutralization), which is reliably quantified by ^13^C NMR spectroscopy [42]. For PS-based NIEs, the surface charge, surface-to-volume ratio, and electric potential have been estimated by calculation but without experimental verification [31]. When the electrophoretic mobility of the nanoparticles is of interest, the measurements can be performed using a standard Zetasizer instrument [39].

## 4. Applications

### 4.1. Solid-Phase Extraction

Sample cleanup and enrichment by SPE has been the major application domain of functionalized nanomaterials from their advent to modern-day analysis [59,60], and this trend includes NIEs. Sulfonated cation-exchange nanotubes, which are made of organosilylated halloysite [61], were recently tested for their performance in the selective SPE of toxic pyrrolizidine alkaloids as alternative candidates to polymeric resins [43,44]. Their practical applicability was demonstrated for LC-MS/MS analysis of spiked honey samples using an optimized protocol, detailed in Figure 3. Importantly, this NIE showed excellent reusability performance with no decrease in recovery or sorbent depletion over six SPE cycles.

However, it appears that the cited work of Prof. Bonn’s group is the only example of using the NIEs in dynamic SPE mode. Other published reports deal with the adsorption of target analytes evaluated in a tedious batch system. Polypyrrole displays high adsorption capacity with respect to the chromate ion [62], which is boosted and becomes more selective after grafting its positively charged nanoparticles onto functionalized silica gel [12]. A group of Iranian researchers proposed using a nano-sized ion-exchange bioresin to remove heavy metal ions from aqueous solutions [55,56,57]. It should be noted that these and other authors point out the possible application of NIEs for environmental remediation, taking advantage of their high metal reactivity and surface area [24]. However, no studies on metal detoxification of environmental and industrial waters could be traced in the literature. 

Concerning bioanalytical applications, a positively charged PMMA-based NIE encapsulated together with magnetite nanoparticles in polymer beads was designed for selective protein binding [11]. At first sight, these composite particles appeared to be encouraging due to their superparamagnetic properties, but, surprisingly, their advantages for magnetic separation were not demonstrated. Guo et al. [23] and Li and coworkers [63] found that commercial superparamagnetic cation or anion NIEs may selectively capture some bacteria by the mechanism of aggregation on the bacterial surface due to electrostatic forces. Such capturing can be used for magnet-guided separation of target bacteria for subsequent MALDI-MS analysis. Another recent application of NIEs incorporating magnetite is the separation of thrombin from clinical serum samples with the help of a magnetic fluidized system [64].

### 4.2. Separation Methods

Ion-exchange materials have a long employment history as a separation-mediated component in chromatography and electrophoresis. Still, some researchers believe that nanoscale dimensionality would give a new lease of life to such mature techniques as IC or CE. The idea of modifying the µm-sized particles of common IC column loadings with a suspension of sub-microparticles is almost as old as the method itself [65,66], but implementation of NIEs as modifiers has not been a prominent response. In an attempt to fill this gap, Dolgonosov et al. [30] constructed a cation-exchange column in which the cation NIE is fixed in the pores of an anion-exchange matrix (Figure 4). Due to the chemical stability and stable sorption properties, such a design may have an advantage over other multifunctional stationary phases, e.g., those based on silica gel [67]. However, no practical applications have so far been reported besides the separation of a model mixture of alkali metal cations and ammonium [31].

Hardware modification is also a mainstream approach in electromigration techniques, particularly for separating basic and/or hydrophobic analytes. Dzema et al. [28] and Polikarpova and coworkers [33] explored the possibility of dynamic coating of fused-silica capillaries using a PS-based NIE functionalized with sulfo groups. When added to the capillary electrolyte, the NIE was shown to form a loose uniform monolayer on the capillary surface which, for certain analytes, made the resolution and separation efficiency better than in the case of untreated capillaries. However, the authors provided no detailed comparison with other cationic capillary modifiers, and some separations (e.g., of inorganic anions [28]) were significantly inferior to the golden CE standard (over 30 separands in one run [68,69]). As a practical application of CE on NIE-modified capillaries, the determination of a number of catecholamines and amino acids [33] or a few inorganic anions [28] in human urine was demonstrated. In another report by the same research group [29], a similar but positively charged NIE was proven as a feasible pseudostationary phase for CEC, another powerful electromigration technique for the separation of a variety of analytes [70]. To cope with the modest durability of NIE coatings, a layer-by-layer approach was recently proposed [34] in which the capillary inner wall is sequentially coated with two layers of oppositely charged NIEs. Due to the benefits of high IEC, large surface-to-volume ratio, and a high affinity toward the fused silica surface, a wider implementation of NIEs in CEC holds promise. However, significant fabrication challenges limit the use of NIEs with CEC, at least in part, and, as such, it has only been reported twice at the time of this writing. Earlier applications of other types of polymeric nanoparticles such as pseudostationary phases for CEC were reviewed by Palmer and McCarney [71].

### 4.3. Miscellaneous Analytical Applications

PS-NIEs also find application in boosting the analytical signal in certain spectroscopic methods. When a cation NIE in the H-form was recently added to the analyzed solution, an unusual (and still inexplicable) effect on the response of various metals in ICP-OES was discovered [32]. This greatly enhanced emission resulted in an improvement in the limits of detection by up to one order of magnitude. In contrast, the use of NIEs in luminescence analysis has a solid foundation as their diluted suspensions possess a marked luminescence. Heavy metal ions, e.g., Cu(II) [27], tend to be concentrated on NIEs and simultaneously quench their luminescence, which affords an indirect method of metal quantification (albeit within a limited concentration range and with ambiguous selectivity). 

Shkinev et al. [35] designed an instrumental setup for the electrochemical determination of acetylcholine in which a cation NIE was used as a modifier of the pores of track membranes. The explicit role of the NIE was to stabilize the interface between two immiscible electrolytes, filling the membrane pores, and to facilitate reversible charge transfer. While the proposed voltammetric sensor demonstrated acceptable analytical characteristics in comparison with other electrochemical methods, its sensitivity calls for improvement, and the applicability to real-world samples awaits examination. 

### 4.4. Toward Industrial Use

A range of NIEs have the potential to be employed in industrial technology. Similarly to those of other engineered nanostructures [72,73], promising applications of NIEs may be segmented into energy, electronics, chemistry, healthcare, and other end-user industries. One of the prospective directions in finding new energy sources is the development of fuel cells, the functioning of which is defined by the proton exchange between the anode and cathode sides of polymer electrolyte membranes [74]. Novel sulfonic acid-functionalized chitin nanowhiskers were recently assessed for the production of direct methanol fuel cell membranes [40]. The results showed that the modification with sulfonic acid groups as the proton-conducting sites enhanced the performance of the manufactured membranes in terms of the proton-conductivity-to-methanol permeability ratio. Similarly, functionalized halloysite nanotubes were examined as acid catalysts in the esterification of a mixture of free fatty acids as a hybrid feedstock model for biodiesel production [53]. Almost 100% feedstock conversion and reusability are attractive attributes of this new nanocatalyst, especially in terms of making the biodiesel production process more efficient and economical. In addition, in the area of developing next-generation electronic devices, NIEs originating from conducting or semiconducting polymers possess (high) conductivity and thus may play a role in advancing research in soft electronics. This application was highlighted in a recent review paper by Wang et al. [75], in which progress in engineering conducting polymers with various nanostructures to be used in soft electronic devices was summarized. 

Nanoscale polypyrrole-based material, containing polystyrenesulfonate as a counter-ion and exhibiting cation-exchange behavior, has been shown to be promising in technological applications [76]. Depending on its morphology, either particles or films, this type of NIE can be employed as a support for a nano-dispersed Pt catalyst used in a hydrogen and methanol oxidation reaction or for electrochemical water softening. Binding of catalytically active rhodium complexes to polycationic polymer NIEs of different sizes via multiplying sulfonated ligands was investigated by Mecking et al. [77] with the goal of combining the advantages of classical homogeneous and heterogeneous catalysts. An ionic polymeric NIE was used to fabricate ultrafiltration membranes with improved water permeability, antifouling nature, and surface hydrophilicity, mainly due to the presence of quaternary ammonium groups [45]. Initial testing revealed that the fabricated membranes improved the separation of oil/water emulsions. Negatively charged polymer nanoparticles were used to modify the rheological properties of fresh cement pastes [41]. High adsorption of NIE on cement produced electrostatic repulsion and steric hindrance between cement grains and, therefore, greatly advanced its yield stress and plastic viscosity. In another application, Sun et al. constructed a conductive textile-based wearable pressure sensor featuring a robust polydopamine–carboxylated carbon tube configuration and capable of detecting a pulse and throat movement or other human motions [78].

In summary, however multifaceted and promising the development of NIEs for industrial use may be, their practical application still remains an elusive goal. More efforts for the commercialization of technical inventions in which these materials play a key role are anticipated.

### 4.5. Drug Delivery

Nanomedicine is arguably the most rapidly emerging application of nanotechnology, and one of its pillars, controlled or targeted drug delivery, has received a great deal of interest from oncologists and medicinal chemists [79,80]. Over the past decade, a wide variety of drug carriers have been synthesized from different types of nanomaterials, of organic, inorganic, or biological nature, and their composites [81], as well as synthetic and natural polymeric nanosystems [82,83,84]. This does not impede, however, the discovery of alternative nanoscale vehicles. A proof-of-concept study by Kuznetsova and her colleagues [37] indicated that NIEs possess some of the features required of an intelligent drug carrier. Using cisplatin as a test drug, it was demonstrated that a cation PS-based NIE is able to load a high drug dose, protect the payload against the human serum environment, and release the drug in therapeutically effective quantities. As can be seen in Figure 5, the amount of platinum discharged in a simulated cancer cytosol differed greatly from that released under normal cytosol conditions, which is a sign of cell-selective drug delivery. Nonetheless, the proposed nanocarrier clearly requires further development to accomplish targeted delivery. Its magnetization [85], e.g., by fabricating nanocomposite particles [9], modification with explicit moieties overexpressed by cancer cells [86], or noncovalent decoration with antibodies [87], might address the issue of enhancing tumor-specific uptake.

The same NIE offers promise as a nanoplatform for the delivery of another first-line chemotherapeutic agent, doxorubicin [36]. The drug loaded on the nanomaterial possesses a three to six times greater cytotoxicity than the same dose of doxorubicin alone. The authors ascribed this effect to the drug enrichment upon attachment to the NIE and expressed their hope that this NIE could find application in clinical practice. Recently, it was shown that the intracellular delivery of doxorubicin depends on the core architecture of pH-responsive polymer nanoparticles, with the loaded particles with a linear core displaying higher toxicity in MCF-7 cells compared to those with a star core [88]. However, an interested reader should not be misguided. High cell-killing ability is not the most important characteristic of an anticancer drug, especially when the cytotoxicity is tested in a simplistic way, i.e., by adding a drug formulation directly to a cell culture without taking into consideration the numerous potential chemical transformations on the way to the cancer cell.

In another recent study, an anionic PMMA-based polymer approved by the FDA was applied to poly(lactic acid) nanoparticles, and, simultaneously, a small-molecule therapeutic drug was incorporated into the structure, thereby creating an effective oral nanoformulation [42]. The drug release from the nanoparticle core largely depends on the pH responsiveness of the polymer coating provided by the presence of carboxyl groups. While it has been explored only with a single antimalarial drug, lumefantrine, this combined nanomaterial is believed to enable the encapsulation of a broad spectrum of hydrophobic active pharmaceutical ingredients. Perhaps the most comprehensive study on drug-carrying systems based on NIEs was performed by Zhang et al. [89]. The authors functionalized monodispersed silica nanoparticles with polydopamine, loaded the resultant material with cyclophosphamide, a therapeutic drug against diffusive alveolar hemorrhage, and proved drug release through photothermal conversion and low pH. Furthermore, good biocompatibility of the nanoformulation and its ability to alleviate disease progression after three weeks of systematic administration were demonstrated in animal experiments. 

### 4.6. Delivery of Contrast Agents 

The biomedical functions of polymeric nanoparticles extend beyond drug delivery [90] as they can also be used as nanocarriers of contrast agents, e.g., for fluorescence bioimaging [91,92]. At least one family of NIEs, based on three types of polymers, including two biodegradable polyesters (see Table 1), and functionalized with negatively charged carboxylate or sulfonate or positively charged trimethylammonium groups, was prepared and investigated for such an application [26]. A fit-for-purpose feature of these NIEs is the nearly quantitative encapsulation of organic dyes, which generates a fluorescence nanomarker that is more than 10 times brighter than typical commercial quantum dots (Figure 6). No release of the dye was observed in a simulated biological medium, PBS with 10% serum, and, even though these are not real blood conditions, the developed NIEs may expect a drive towards being adapted in medical diagnostics.

In conclusion, different types of NIEs are pursued for the development of novel analytical, technological, and medicinal tools and materials. Their benefits and current limitations are summarized in Table 2.

## 5. Conclusions

The past few years have seen great efforts toward putting NIEs on the map of advanced engineered nanostructures. Tailor-made fabrication strategies are continuously being developed to improve the nanoscale characteristics of NIE materials. While the approaches can differ, those which use versatile direct polymerization hold greater promise in terms of required cost efficiency, end-product variability, and quality metrics. On the other hand, post-polymerization based on nanoprecipitation is an attractive approach to induce polymer nanoization via a simple charge- and kinetically controlled process. More problematic is the precise control over the NIE nanostructure from the standpoint of its ion-exchange behavior, which remains a bottleneck for approved synthesis qualification. Advanced characterization techniques that can address this challenge are therefore urgently required. In addition, the issues of environmentally friendly and large-scale manufacturing have not received due attention. In this regard, materials with massive natural occurrence, such as chitosan [93] or nanoclays [94], seem to present green and sustainable alternatives to the usual NIEs based on fossil-derived polymeric substrates. 

As reflected in this review, a representative number of NIEs have been explored with respect to increasing useful applications. However, most of the research endeavors looked like ‘one-bullet shootings’, with a single study or few studies devoted to a specific application, and, therefore, remain at the level of qualitative research. In the field of analytical chemistry, nanoparticulation is clearly gaining popularity for IEPs. Still, as with pursuing every novel analytical approach or tool, analysts should be guided by the principle of proving the advantage of an NIE-mediated system over the existing analytical methodology in order to justify their time, effort, and costs. To bridge the gap between the laboratory bench and industry, extensive collaborative investigations from multiple disciplines are needed. In particular, the long-term reliability of devices and units fabricated from NIEs should be thoroughly assessed. The functionalities, optimized performance, and user friendliness of current NIE-based designs are also limited, demanding a devoted research response. Finally, the path of NIEs from concept to clinical translation is by no means straightforward. Using NIEs made from biocompatible and degradable polymers would be the most promising roadmap strategy to ensure their biosafety and clearance from living organisms. In addition, NIEs designed as nanocarrier vectors require in-depth quality control before loading to produce drug formulations free of impurities, as well as the assessment of biocompatibility, immunogenicity, and possible side effects prior to testing in humans.

## Figures and Tables

**Figure 1 molecules-28-06490-f001:**
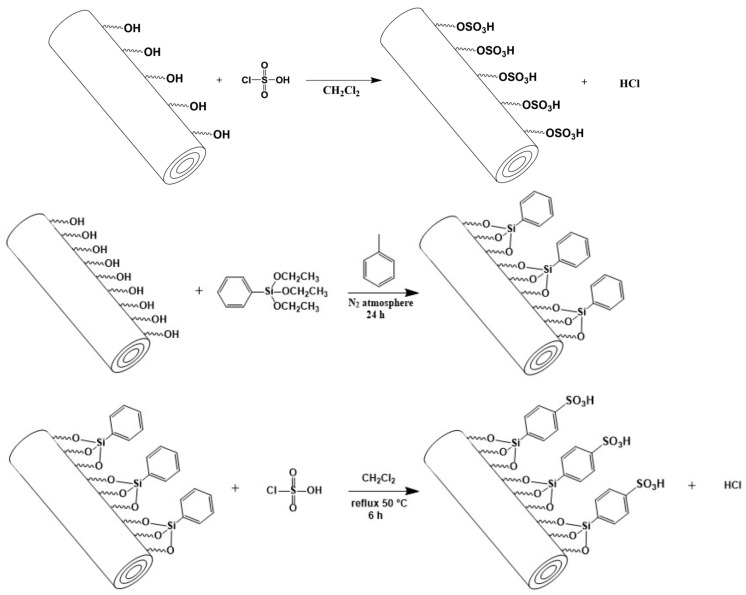
Synthesis of halloysite-based NIEs consisting of direct sulfonation [43] or organosilylation and sulfonation steps [44].

**Figure 2 molecules-28-06490-f002:**
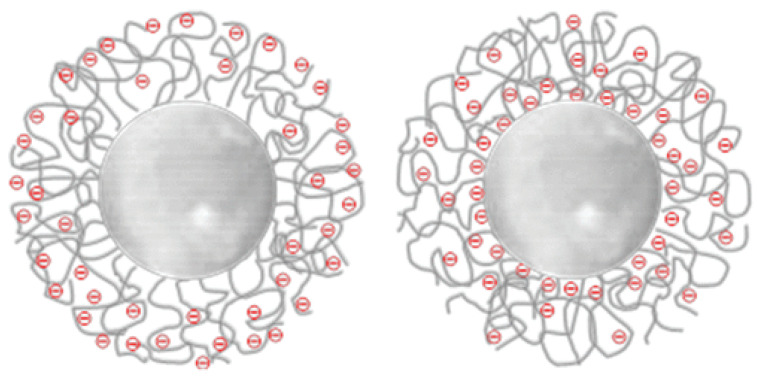
Charge distribution in core–shell NIEs functionalized with acrylic and methacrylic acid (left- and right-hand cartoon, respectively) [24].

**Figure 3 molecules-28-06490-f003:**
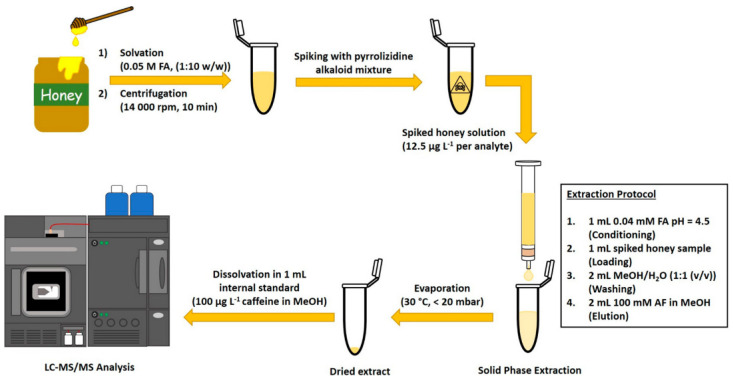
Experimental workflow of LC-MS/MS of pyrrolizidine alkaloids in honey matrix using sulfonated halloysite nanotubes as material for SPE [44].

**Figure 4 molecules-28-06490-f004:**
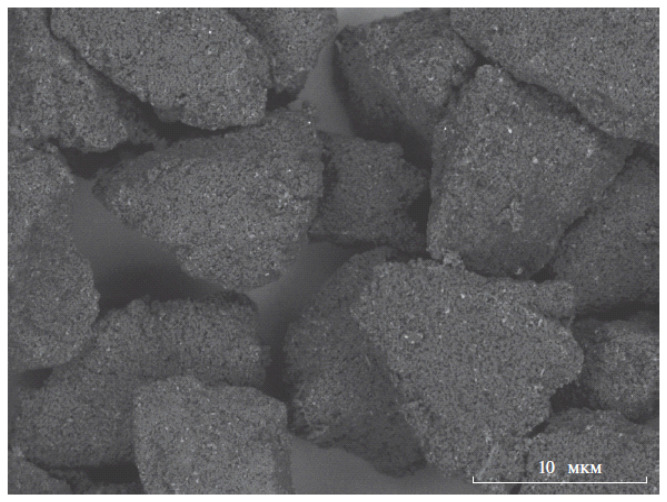
TEM image of NIE-modified column material. Cation NIE particles are seen as light inclusions. The scale marker shows a 10 µm range [30].

**Figure 5 molecules-28-06490-f005:**
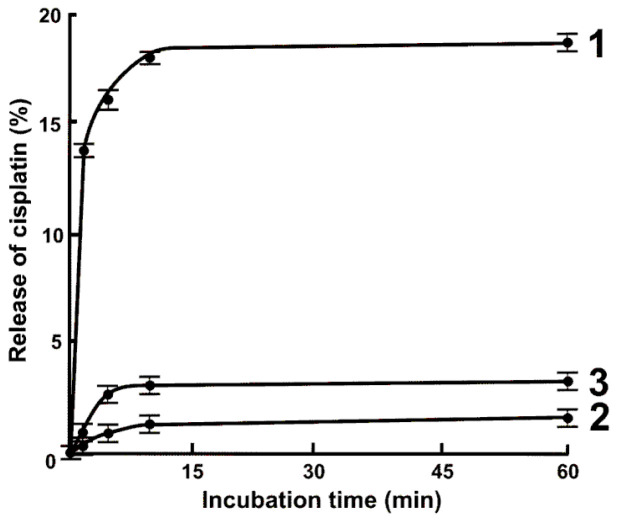
In vitro release of cisplatin from the cation NIE under the action of simulated cancer (**1**) and normal (**3**) cytosol. For comparison, trace (**2**) shows the release behavior in cancer-like physiological buffer containing no active cytosolic components [37].

**Figure 6 molecules-28-06490-f006:**
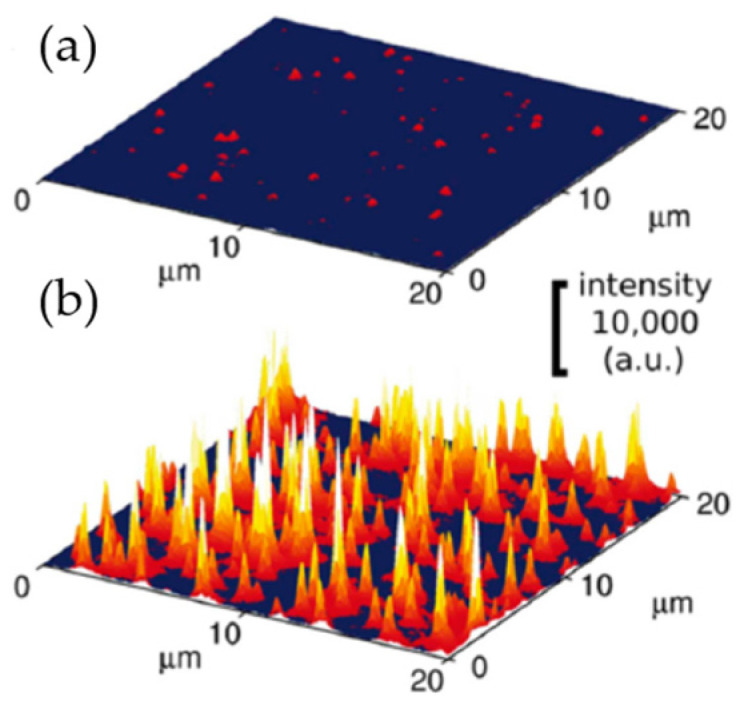
Fluorescence microscopy images of (**a**) quantum dots (Qdot 585) and (**b**) PMMA-SO_3_H nanoparticles loaded with a cationic dye of rhodamine class [26].

**Table 1 molecules-28-06490-t001:** Different types and applications of NIEs ^a^.

Matrix	Ionizable Groups	Fabrication	Shape/Size (nm)	Properties	Application	Ref.
PS	Sulfonic acid groups (SG)	Emulsion or emulsifier-free polymerization	Spherical/40–90	IEC, 0.7–2.2 meq g^−1^	-	[21]
PS	Trimethylamine groups	Emulsion polymerization followed by surface functionalization	Spherical/150	IEC, 2.35 mmol g^−1^	Binding of BSA (after the preparation of magnetic/ion-exchange composite beads)	[11]
Polystyrene modified with sodium styrene sulfonate	SG	Emulsion polymerization followed by surface modification by a two-step polymerization reaction	Spherical/30–80	Surface charge density, 5.9–27.7 µC cm^−2^	-	[22]
Polypyrrole grafted onto functionalized silica gel	Pyrrolium groups	Micelle technique	Opal-like/100	IEC, 1.78 meq g^−1^	Adsorption of Cr(VI)	[12]
Poly-d,l-aspartic acid	Carboxylic groups	Commercial product (Chemicell GmbH, Berlin, Germany)	Spherical/100	-	Capturing of bacteria (after coating on magnetite nanoparticles)	[23]
PMMA covered with poly(*N*-isopropylacrylamide)	Carboxylic groups	Two-stage emulsion polymerization	Spherical/90–260	-	-	[24]
Chitosan	Tripolyphosphate groups	Ionic gelation	Nonspherical/105–209	-	-	[25]
PMMA, poly(d,l-lactide-co-glycolide) or polylcaprolactone	Carboxylate, sulfonate or trimethylammonium groups	Functionalization and nanoprecipitation	Spherical/15	-	Loading of a fluorescent contrast agent (after postmodification by surfactants)	[26]
PS	SG or trimethylamine groups	Grinding of commercial cation or anion exchanger	Nonspherical/50–300	IEC, 0.1 and 0.95 meq mL^−1^ for cation and anion NIE, respectively	Luminescence analysis of metals	[27]
					CE of inorganic anions (in urine)	[28]
					CEC of carboxylic acids (in wine)	[29]
					IC of alkali metals and ammonium	[30,31]
					ICP-OES of metals	[32]
					CE of catecholamines and amino acids (in urine)	[33]
					CEC of catecholamines and amino acids	[34]
					CEC of carboxylic acids (in wine)	[35]
					Loading of an anticancer drug (doxorubicin)	[36]
					Loading of an anticancer drug (cisplatin)	[37]
Aliphatic polyester	SG	Polyesterification	Platelet/400	-	-	[38]
Poly(stearyl methacrylate)– poly(benzyl methacrylate) with incorporated 2-((methacryloyloxy)ethyl)-trimethylammonium	Trimethylamine groups	Polymerization-induced self-assembly	Spherical and nonspherical (worms, vesicles)	-	-	[39]
Chitosan	SG	Extraction by HCl (from shell chitin), dialysis, freeze-drying, modification by propane- 1,3-sultone	Whiskers/diameter, 15–30; length, 150–300	IEC, 0.60–0.91 mmol g^−1^	Preparation of nanocomposite polymer membranes for direct methanol fuel cell	[40]
Copolymers of styrene, acrylic acid, N-dimethyl acryl amide, and methyl allyl polyoxyethylene ether	Carboxylic groups	Precipitation polymerization	Nonsperical/7–146	Charge density, 0.09–1.5 meq g^−1^	Modification of the rheological properties of cement pastes	[41]
Poly(lactic acid) covered with methyl methacrylate polymer	Carboxylic groups	Flash nanoprecipitation	Spherical/59–454	Surface charge (no data)	Loading of antimalarial drug (lumefantrine)	[42]
Halloysite	SG	Direct sulfonation or organosilylation and sulfonation	Tubes	-	SPE of pyrrolizidine alkaloids	[43,44]
Poly(2-(dimethylamino) ethyl methacrylate-co-4-vinylbenzyl chloride)	Quaternary ammonium groups	Quarterization precipitation polymerization	Spherical/50–80	-	Oil/water emulsion separation (after the incorporation into polysulfone ultrafiltration membranes)	[45]

^a^ CE = capillary electrophoresis; CEC = capillary electrochromatography; IC = ion chromatography; ICP-OES = inductively coupled plasma optical emission spectroscopy; IEC = ion-exchange capacity; SPE = solid-phase extraction.

**Table 2 molecules-28-06490-t002:** Summary of the main advantages and disadvantages of NIEs.

Advantages	Disadvantages
Large specific surface area High IEC Surface-to-volume area Reactivity Stability in suspension Rational design strategies Solid preparation technology Reusability Wealth of analytically and technically attractive properties	Outdated methodology for assessing ion-exchange properties Shortage of large-scale manufacturing approaches No real-world applications for environmental remediation Uneasiness of NIE-based devices and unitsMarginal biosafety Lack of targeted delivery function

## Data Availability

Not applicable.

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
