# Peer review of "Nanoscale Ion-Exchange Materials: From Analytical Chemistry to Industrial and Biomedical Applications"

_molecules, 2023, doi:10.3390/molecules28186490_

Round 1

Reviewer 1 Report (New Reviewer)

The review article titled "Nano-Scale Ion-Exchange Materials: Exploring Analytical Chemistry to Industrial and Biomedical Applications" presents a comprehensive overview of various Nano-Scale Ion-Exchange (NIE) materials, detailing their fabrication techniques and ion exchange behaviors. The article delves into their wide-ranging applications. However, there are a few aspects that require attention and correction:

1.       The introductory section should encompass a thorough discussion of relevant information, pertinent statistics, and a comprehensive review of the existing literature.

2.       The quality of the figures included in the article is subpar, and some could potentially be better represented if the authors took on their creation.

3.       Given its nature as a review article, employing a single figure might not be optimal. It's recommended to explore options like using combined figures or merging the existing single figures to enhance clarity.

4.       To enhance clarity and organization, the various strategies utilized to synthesize NIEs should be systematically categorized under separate subheadings.

5.       The Conclusion and prospectives section should concisely summarize the advantages and limitations of NIEs using bullet points for enhanced readability.

1-      English language should be revised.

Minor revision

Author Response

  1. The introductory section should encompass a thorough discussion of relevant information, pertinent statistics, and a comprehensive review of the existing literature.

To our way of thinking, ‘a comprehensive review of the existing literature’ is the matter of the whole review, while the subject of nano-sized ion-exchangers received sufficient introductory coverage. Perhaps the reviewer could give us a clue what other relevant information is necessary.

  1. The quality of the figures included in the article is subpar, and some could potentially be better represented if the authors took on their creation.

To our regret, we have no possibility to create better figures as they are taken from original literature.

  1. Given its nature as a review article, employing a single figure might not be optimal. It's recommended to explore options like using combined figures or merging the existing single figures to enhance clarity.

We failed to understand what the reviewer means here by ‘a single figure.’ If one figure per section, this is not what we have in the text.

  1. To enhance clarity and organization, the various strategies utilized to synthesize NIEs should be systematically categorized under separate subheadings.

Done,

  1. The Conclusion and prospectives section should concisely summarize the advantages and limitations of NIEs using bullet points for enhanced readability.

We like this idea. However, after carefully checking recent publications, we found out that this style is not accustomed in Molecules. Therefore, we have preferred to include a separate table showing the advantages and limitations of NIEs.

6-      English language should be revised.

The re-submitted version was revised by a native speaker, PhD in chemistry, and additionally run it through the Grammarly program (professional version). If there are still some language imperfections, we would be happy to learn them.

Reviewer 2 Report (New Reviewer)

Overall, it is an interesting review on ion-exchange materials and their potential applications. But the following points should be addressed prior to reconsideration.

(1) More recent research cases need to be introduced in the introduction. E.g., https://doi.org/10.1007/s11356-020-10162-y, https://doi.org/10.1016/j.jhazmat.2016.12.003

(2) “3. Characterization” This part should complement the description. An explanation of the meaning of each characteristic and a table comparing the exchange capacities seem necessary.

(3) “NIEs for environmental remediation” This part should be discussed in a separate chapter.

(4) “chitosan [90] or nanoclays [91], seem to present green and sustainable alternatives to usual NIEs based on fossil-derived polymeric substrates.” This part needs to be dealt with in more depth in the main manuscript or in a separate chapter.

Author Response

We are grateful to the reviewer for this comment. The mentioned publications, as well as another recent paper (ref. 52), are included in the revised manuscript.

(2) “3. Characterization” This part should complement the description. An explanation of the meaning of each characteristic and a table comparing the exchange capacities seem necessary.

We added explanations to less common characteristics. As to the ion-exchange capacities, all available data are shown in Table 1.

(3) “NIEs for environmental remediation” This part should be discussed in a separate chapter.

This would be a handsome supplement to the review. However, as mentioned in Section 4.1, we found no practical applications of nano-sized ion-exchangers for environmental remediation.

(4) “chitosan [90] or nanoclays [91], seem to present green and sustainable alternatives to usual NIEs based on fossil-derived polymeric substrates.” This part needs to be dealt with in more depth in the main manuscript or in a separate chapter.

We would prefer to devote a separate review paper to these materials as soon as a wealth of data on their applications will be accessible. In this issue, only synthetic NIEs are in focus (see Introduction).

Round 2

Reviewer 1 Report (New Reviewer)

The authors did not address all my comments yet.

Poor figure quality could be resolved by redesigning a representative figure by the authors.

Single figures mean you can merge figures together as a,b, ....

Author Response

Poor figure quality could be resolved by redesigning a representative figure by the authors.

We apologize but such a treatment would hardly be possible within 5 days given to us for a revision. Moreover, some figures are too complicated or represent images (Figures 4 and 6) unsuitable for redesigning. Besides, if we reproduce a figure with permission, are we in a position to change it?

Single figures mean you can merge figures together as a,b, ....

We find this unpractical as Figures 1 and 2 (both on fabrication) each consist of 2-3 subfigures, while the other four figures reflect much different applications of NIEs to be merged.

Reviewer 2 Report (New Reviewer)

I think it can be published in its current form.

Author Response

Thank you for your time and effort to improve the quality of our manuscript. 

This manuscript is a resubmission of an earlier submission. The following is a list of the peer review reports and author responses from that submission.

Round 1

Reviewer 1 Report

This work proposes a brief review on nano-scale ion-exchange materials, and its application in analytical chemistry, industrial and biomedical was introduced. After assessing the whole manuscript, the subject is worthy of investigation. However, some questions should be addressed to improve the quality manuscript. The authors should address the concerns/drawbacks listed below:

Comment 1:The authors should have a more comprehensive understanding of the research achievements in NIEs field and conduct a comprehensive review of NIEs. The references are relatively few, especially a large numbers of recent related studies should be added.

Cmment 2: Although the author's topic selection is novel, but the review must have a deep understanding of the field, and it is obvious that the author does not have a comprehensive summary on the application of nano-scale ion-exchange materials.

Comment 3: It is noted that this manuscript needs careful editing by some expertise in English editing. Please paying particular attention to English grammar, spelling, and sentence structure, so that the manuscript are clear to the reader.

Comment 4: In this part “2. Rational Design and Fabrication Strategies”, different preparation methods should be listed clearly, with representative figures attached. It is obvious that the authors does not summary clearly and comprehensively in this part.

Comment 5: Although the title of the paper is nano-scale ion-exchange materials, most of the materials introduced in the paper are polymer composites materials.

Comment 6: The figures in the manuscript are too few and the pictures are very unclear, and the format of the figures is very irregular. The authors should add more tables and clear pictures in the manuscript.
Comment 7: The format of the references in this manuscript does not meet the requirements of the journal. Before submitting the manuscript, authors should carefully check the format of the references. If you are unsure, please consult the formatting instructions to authors that are given under the instructions. 

I regret that your manuscript in the present state cannot be published in Molecules. Once the above concerns are fully addressed, the manuscript could be resubmit in this journal.

Reviewer 2 Report

The manuscript titled as "Nano-scale ion-exchange materials: From analytical chemistry to industrial and biomedical applications" provides a brief review about the emerging landscape of various NIE materials and summarizes their actual and potential applications. It is meaningful for the development of NIE. And the manuscript is well organized. The submission can be accepted on Molecules after major revision. Here are some suggestions.

1. What are the differences of AIE fabricated by grafting ionic functional groups onto the matrix with different strategies?

2. Please give the advantages and disadvantages of different strategies for the synthesis of NIEs with desired nanoscale characteristics.

3. In page 8, line 114,why some researchers give preference to more tedious synthetic schemes?

4. Compared with other nanomaterials, what are the unique advantages of NIE applied to solid phase extraction, separation methods, miscellaneous analytical applications, etc.

5. In page 14, line 347-348, How can NIE quantify the encapsulation of organic dyes? What is the interaction between NIE and organic dyes?

6.What is the main challenge for the precise control of NIE nanostructures? Whether the authors can give some suggestions about the strategy?

7. Some formatting errors should be carefully checked and corrected

I would suggest some English polishing so it could become publishable.

Round 2

Reviewer 1 Report

Before submitting a revision, the author should be sure that your manuscript has been properly prepared and formatted. The figures in the paper are really fuzzy, and the pictures selected are not representative. Besides, why does the conclusion also cite references? The paper still needs extensive revision and is not currently available for publication in the MOLECULES.

Please paying particular attention to sentence structure, so that the manuscript are clear to the reader.